# CO-LEARNING SYNAPTIC DELAYS, WEIGHTS AND ADAPTATION IN SPIKING NEURAL NETWORKS

## ABSTRACT

Spiking neural networks (SNN) distinguish themselves from artificial neural networks (ANN) because of their inherent temporal processing and spike-based computations, enabling a power-efficient implementation in neuromorphic hardware. In this paper, we demonstrate that data processing with spiking neurons can be enhanced by co-learning the connection weights with two other biologically inspired neuronal features: 1) a set of parameters describing neuronal adaptation processes and 2) synaptic propagation delays. The former allows the spiking neuron to learn how to specifically react to incoming spikes based on its past. The trained adaptation parameters result in neuronal heterogeneity, which is found in the brain and also leads to a greater variety in available spike patterns. The latter enables to learn to explicitly correlate patterns that are temporally distanced. Synaptic delays reflect the time an action potential requires to travel from one neuron to another. We show that each of the co-learned features separately leads to an improvement over the baseline SNN and that the combination of both leads to state-of-the-art SNN results on all speech recognition datasets investigated with a simple 2-hidden layer feed-forward network. Our SNN outperforms the ANN on the neuromorpic datasets (Spiking Heidelberg Digits and Spiking Speech Commands), even with fewer trainable parameters. On the 35-class Google Speech Commands dataset, our SNN also outperforms a GRU of similar size. Our work presents brain-inspired improvements to SNN that enable them to excel over an equivalent ANN of similar size on tasks with rich temporal dynamics.

## 1 INTRODUCTION

Spiking neural networks (SNN), seen as the third generation of neural network models (Maass, 1997), have recently attracted growing attention as a low-power alternative for artificial neural networks (ANN). Unlike ANN implementations (García-Martín et al., 2019), SNN can enable power-efficient processing on specific neuromorphic hardware such as SENeCa (Yousefzadeh et al., 2022), the Intel Loihi2 (Orchard et al., 2021) or IBM TrueNorth (DeBole et al., 2019). The main power gains can be attributed to SNN inherent event-based computations (by means of spikes) and sparsity, reducing the number of multiplications that are required. A recent study (Stöckl & Maass, 2021) illustrated these potential SNN energy savings.

Recent years have shown great progress in learning algorithms for deep SNN. Especially training SNN based on backpropagation-through-time with surrogate gradients (Neftci et al., 2019), helped overcome the problem of the non-differentiability, introduced by thresholding mechanism in a spiking neuron. These advances enabled SNN to move to deeper and more complicated model architectures using attention mechanisms (Yao et al., 2023) or transformers (Zhou et al., 2023) and (Zhu et al., 2023). The main issue related to SNN however persists: frequently the SNN model does not perform as well as the equivalent ANN.

In biology, researchers have found that the brain is equipped with a plethora of powers to process spike trains adequately. One of those is the axonal delay. The transmission speed of an action potential is known to depend on the myelination of the axon and thus determine how a spike is delayed (Purves et al., 2001). Furthermore, these delays are crucial in sensory processing (Orchard & Etienne-Cummings, 2014) and known to adapt during the learning process (Lin & Faber, 2002). A delay-enabled spiking network was also found to be able to compute a richer class of functions

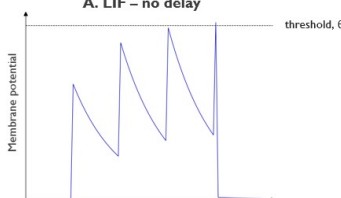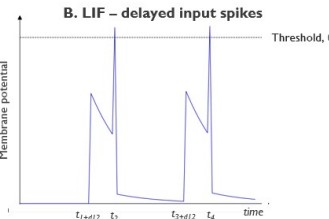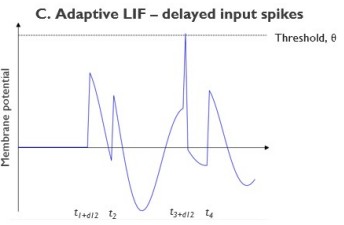

Figure 1: Illustration of the variety in responses for different neuron models: Two input spike trains are processed by three different neurons with equal weights. Input neuron 1 spikes at $(t_2, t_4)$ and input neuron 2 spikes at $(t_1, t_3)$. For all neurons we show the evolution of the membrane potential over time in response to these input spike trains: **(a)** a typical leaky-integrate-and-fire (LIF) neuron, spikes at the timestep of the last incoming spike, $t_4$. **(b)** a LIF neuron with delayed input spikes from input neuron 2, produces spikes at timesteps $t_2$ and $t_4$. **(c)** an adaptive neuron processes the same delayed input spikes from input neuron 2 and only produces a spike at timestep $t_3 + d_{23}$, the latency of the spikes coming from neuron 2.

than a threshold circuit with adjustable weights (Maass & Schmitt, 1999). Another differentiator between classical ANN and the brain is the widespread heterogeneity and neuronal adaptation processes taking place. Neurons of all forms and shapes are found, each enabling a wide array processing functions (Gerstner & Kistler, 2002). Moreover, biological neurons exhibit slow dynamical processes that act at longer time scales, enabling processing of events that are temporally distanced in an implicit manner. These adaptive processes often limit the number of spikes produced.

Combining the ability to optimize the weights, delays and training neuronal parameters can lead to a more diverse and possibly improved internal representation. Figure 1 shows the responses of A) a typical leaky integrate-and-fire (LIF) neuron, B) a neuron with delayed input spike trains and C) a neuron with delayed input spike trains and neuronal adaptation, for a spike train of four equidistant input spikes and all equal connection weights. It can clearly be seen that the output spikes patterns are vastly different. The LIF neuron (A) will always respond the same while the others provide a wider range of possible results because of the trainable delay (B) and non-linear trainable adaptation processes (C). Both extensions show to be complementary in providing further memory. The axonal delay allows the neuron to explicitly correlate incoming spikes at longer timescales whereas the adaptation implicitly alters a neurons behavior based on its past regime.

In this study, we present a SNN with adaptive neurons and synaptic delays that are co-optimized. Whereas normally, as in ANN, just the synaptic weights are trained, we show that co-learning the delays and adaptation parameters, individually enhance the performance of the SNN model, and that the combination of them also leads to state-of-the-art SNN results. The contribution of this study is as follows:

- We present a novel SNN model in which both synaptic weights and delays are co-optimized in collaboration with the neuronal adaptation parameters.
- We analyze the biologically plausible neuronal adaptation parameters and which effects parameter boundaries have on the neurons working regime and on the SNN performance on three speech recognition datasets.
- We show that inclusion of these more complex neurons through adaptation and the addition of trainable delays for every synapse specifically leads to state-of-the-art results for spiking neural networks. The proposed SNN even outperforms its non-spiking counterpart with equivalent model size on the speech recognition problems.

## 2 RELATED WORK

**Learning algorithms for spiking neural networks.** Recently, there has been a rapid evolution in the development and progress of SNN learning paradigms. In general, either a trained ANN is converted into a rate-based SNN, aiming at minimal performance losses due to the ANN-SNN conversion (Deng & Gu, 2021), (Bu et al., 2022) or the SNN is trained directly as a spiking neural network. A directly trained SNN is typically trained with backpropagation-through-time with surro-

gate gradients (Neftci et al., 2019), (Zenke & Vogels, 2021). These surrogates are used to approach the derivative of non-differential Heaviside function, introduced by the spiking mechanism. A recent study went one step further in using differentiable spikes (Li et al., 2021) for temporal credit assignment. Moreover, the spike-element-wise (SEW) ResNet (Fang et al., 2021a) was proposed for training SNN without the vanishing/exploding problem introduced by surrogate gradients, paving the way for deeper SNN models.

**Learning of neuronal parameters and neuronal heterogeneity.** Another trend in SNN research is to learn the optimal distribution of the neuronal (leakage) parameters (Yin et al., 2021), (Fang et al., 2021b) and (Rathi & Roy, 2023). Moreover, the Gated LIF neuron (Yao et al., 2022) was proposed to control the fusion of learnable membrane-related parameters. Additionally, in many works the heterogeneity of neurons in SNN proved to be beneficial for improved recognition (Deckers et al., 2022), (Perez-Nieves et al., 2021) (Chakraborty & Mukhopadhyay, 2023).

Different methods for including neuronal adaptation processes have been proposed. A first class contains neurons with an adaptive threshold, which is increased after every spike and exponentially decays over time (Salaj et al., 2021), (Yin et al., 2021). Others (Falez et al., 2019) proposed a method for tuning the thresholds with specific target spike timestamps as an objective. In other work (Gast et al., 2020), the adaptive thresholds were combined with a synaptic depression model, which showed to replicate both bursting and steady-state behavior. Another adaptation method is based on adaptation currents, coupling a secondary variable to the sub-threshold membrane potential and its spike activity (Brunel et al., 2003). This method was successfully formalized into the AdLIF spiking neuron model (Bittar & Garner, 2022) for SNN and shown to outperform adaptive threshold based models.

**Delays in SNN.** Many methods have been proposed for adapting propagation delays, inspired by spike timing dependent plasticity (Wang et al., 2013) or based on the ReSuMe learning rule (Zhang et al., 2020). A method for training per neuron axonal delays based on the SLAYER learning paradigm was proposed (Shrestha & Orchard, 2018) and extended (Sun et al., 2023) with trainable delay caps. Recently, the effects of axonal synaptic delay learning were studied by pruning multiple delay synapses (Patiño-Saucedo et al., 2023), modeling a one-layer multinomial logistic regression with synaptic delays (Grimaldi & Perrinet, 2023) and learning delays represented trough 1D convolutions with learnable spacings (Hammouamri et al., 2023). Similarly, in order to train synaptic delays, spike trains were transformed into continuous analog, differentiable signals (Wang et al., 2019). Surprisingly, only learning the delays (Grappolini & Subramoney, 2023) showed to achieve comparable performance as only learning the weights.

## 3 SYNAPTIC DELAY-LEARNING IN ADAPTIVE SPIKING NEURAL NETWORKS

A spiking neural network (SNN) is a biologically inspired type of neural network, in which spikes, i.e., binary events are used to communicate between layers of spiking neurons. In this section we elaborate on the fundamental properties of spiking neurons and the methods used in this work to train the synaptic weights, synaptic delays and neuronal parameters in multi-layer networks of spiking neurons.

### 3.1 SPIKING NEURONS

Spiking neurons differ from classical neurons in ANN because of their inherent time-dependent processing of input data. Incoming spikes are multiplied by the synaptic weights and accumulated over time for every neuron. When this neuronal state, i.e., the membrane potential crosses the spiking threshold, a neuron emits a spike to a subsequent layer. The membrane potential is maintained over time and thus creates an internal memory for every individual neuron.

In this paper, we use the adaptive leaky integrate-and-fire neuron (AdLIF) model (Bittar & Garner, 2022) with updated parameter boundaries. To highlight this modification, the neuron model is named AdLIF+. In this model, two internal states are kept: the membrane potential and the adaptation current, which provides the neuronal adaptation. Formally, $u[t]$, $w[t]$, and $s[t]$ respectively represent the membrane potential, the adaptation current and the presence of a spike, at time step t. In the AdLIF neuron model, there are four trainable neuronal parameters: $\alpha$ and $\beta$ denote the leak of u[t] and w[t] respectively, while *a* and *b* describe the characteristics of the adaptation current. The

adaptation current is coupled with the sub-threshold membrane potential by $a$ while $b$ represents the spike-triggered adaptation. A neuron generates a spike at timestep t when the membrane potential crosses the firing threshold, $\theta$, at t. Mathematically, the threshold $\theta$ is represented as a Heaviside-function. The full discrete-time model, with time step size equal to *1ms*, is described in Equation (1). The trainable neuronal parameters are highlighted in bold.

$$
\begin{aligned}
u[t] &= \boldsymbol{\alpha}u[t-1] + (1-\boldsymbol{\alpha})(I[t] - w[t-1]) - \theta s[t-1] \\
w[t] &= \boldsymbol{\beta}w[t-1] + (1-\boldsymbol{\beta})\boldsymbol{a}u[t-1] + \boldsymbol{b}s[t-1] \\
s[t] &= u[t] \geq \theta
\end{aligned}
\tag{1}
$$

The adaptation, implemented by means of this adaptation current $w[t]$ is affected by both the neurons instantaneous membrane potential and a spike-triggered fraction. This contrasts with the classic adaptive neuronal threshold adaptation, in which only the spiking activity is taken into account (Yin et al., 2021). This model was chosen because of its proven superior performance in comparison with a leaky integrate-and-fire (LIF) model with an adaptive neuronal threshold (Bittar & Garner, 2022). The computational graph of the neuron model, rolled out over time, is shown in Figure 2. The yellow box, $w[t]$, represents the addition of the adaptation variable to the classic LIF neuron model and the blue arrows connecting the the internal variables represent the corresponding trainable neuron parameters.

### 3.2 TRAINING MULTI-LAYER SNN

### 3.2.1 MODEL ARCHITECTURE

Similarly to earlier studies, the SNN in this work consists of a simple feed-forward network with two hidden layers. Figure 2 shows the network architecture, rolled out over time. In general, neuron $i$ in hidden layer $l$ receives $I_i^l$, the pre-synaptic current, which consists of two elements, as shown in Equation (2). The feed-forward synapses from the previous layer $l-1$ have associated weights $F_{ji}^{l-1}$ and carry spikes from the same time step $t$ to neuron j and the neuronal bias, $b_i^l$. These input spike trains are summed over all pre-synaptic neurons $j = 1, ..., N_{l-1}$. Spike trains are represented by $s[t] \in [0, 1]$. Typically, in this type of architecture all feed-forward are connected in an all-to-all fashion.

$$
I_l^i = \sum_{j=1}^{N_{l-1}} F_{ji}^{l-1} s_j^{l-1}[t] + b_i^l
\tag{2}
$$

The readout mechanism, which is used to derive the outputs of the SNN, consists of a single layer. This layer consists of neurons with infinite threshold. These neurons have no memory and hence the membrane potential is equal to the the weighted inputs. The output of the SNN model is the sum of the membrane potentials of the output neurons over time, passed though a softmax layer. The outputs are shown in Equation (3).

$$
u_{out} = \sum_t \frac{e^{u_{out}[t]}}{\sum_j e^{u_{out,j}[t]}}
\tag{3}
$$

### 3.2.2 TRAINING PROCEDURE

Typically, spiking neural networks (SNN) are trained via backpropagation-through-time (BPTT) with surrogate gradients (Neftci et al., 2019). In these methods, the summed membrane potentials of the output neurons, see Equation (3), are used to constitute the cross-entropy loss of the network, unrolled over time. The loss with respect to class c for a batch size N is represented as:

$$
\mathcal{L}_c = \frac{1}{N} \sum_{n=1}^N -log(\frac{e^{u_{out,c}}}{\sum_j e^{u_{out,j}}})
\tag{4}
$$

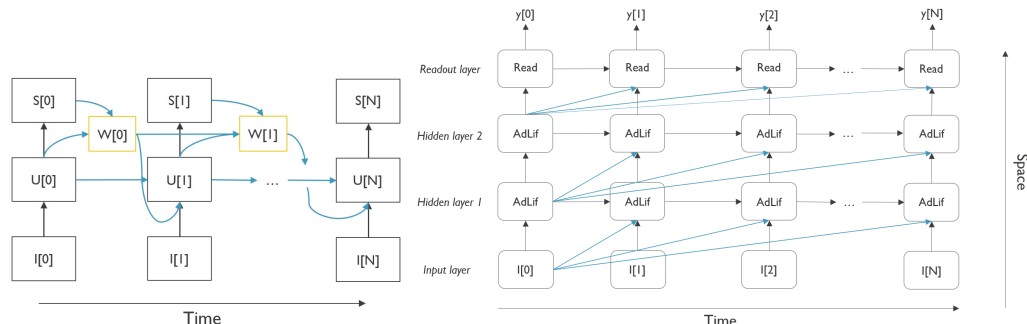

Figure 2: **Left**: Computational graph of the AdLIF neuron model, unrolled over time. The yellow blocks denote the additional adaptation current parameter, which depends on the membrane potential u[t] and the spike activity, s[t] at the previous timestep. **Right**: 2-layer fully connected architecture with readout of the output neurons membrane potential, unrolled over the time. The blue connections denote the potential delays in the network. For clarity the delays starting from time step 1 onwards were omitted.

Based on the chain rule in error-backpropagation, the weight update for neuron $i$ in the penultimate layer $l$ for a sequence of T timesteps is shown in Equation (5).

$$\frac{\delta \mathcal{L}_c}{\delta w^l} = \frac{1}{T} \sum_{t=1}^{T} \sum_{m=0}^{t} \frac{\delta \mathcal{L}_c[t]}{\delta u_{out}[m]} \frac{\delta u_{out}[m]}{\delta s_l[m]} \frac{\delta s_l[m]}{\delta u_l[m]} \frac{\delta u_l[m]}{\delta w_l} \tag{5}$$

In these methods, surrogate gradients are used to approximate the non-differentiability, which was introduced by the thresholding mechanism (Heaviside function), $\frac{\delta s[t]}{\delta u[t]}$, with a differentiable function in the backward pass of error-backpropagation. For simplicity and comparability with previous work, the boxcar surrogate gradient function, shown in Equation (6), was used in this work.

$$\frac{\delta s[t]}{\delta u[t]} = \begin{cases} 0.5 & \text{if } |u[t] - \theta| \leq 0.5 \\ 0 & \text{otherwise} \end{cases} \tag{6}$$

### 3.3 INTRODUCING SYNAPTIC DELAYS

In contrast to ANNs and most other SNN in which connections are defined by a weight parameter, in our work connections between neurons are characterised by two synaptic parameters: a weight and a delay. The addition of synaptic delays leads to an adjusted neuronal processing model. Now, the spike train for neuron $i$ from layer $l$ is characterised by $I_l^i$ in Equation (7). The synaptic delay can be found in $s_k^l[t - d_{ji}]$, where $d_{ji}$ denotes the delayed spikes between neuron j and i.

$$I_l^i = \sum_{j=1}^{N_{l-1}} F_{ji}^{l-1} s_j^{l-1}[t - d_{ji}] + b_i^l \tag{7}$$

Training of the synaptic delays is based on the SLAYER learning method (Shrestha & Orchard, 2018) for training axonal delays, applied to individual synapses. The delay kernel, $\epsilon_d$, is convolved with spike train s[t], to get the delayed spike kernel, $a^l[t] = (\epsilon_d * s^l)[t]$. Similarly to the case in which delays were equal to 0 (i.e. without delays), the derivative of the loss with respect to the synaptic weight and delays are now computed as shown in (8) and (9).

$$\frac{\delta \mathcal{L}_c}{\delta w^l} = \frac{1}{T} \sum_{t=1}^{T} \sum_{m=0}^{t} \frac{\delta \mathcal{L}_c[t]}{\delta u_{out}[m]} \frac{\delta u_{out}[m]}{\delta a_l[m]} \frac{\delta a_l[m]}{\delta u_l[m]} \frac{\delta u_l[m]}{\delta w_l} \tag{8}$$

$$\frac{\delta\mathcal{L}_c}{\delta d^l} = \frac{1}{T}\sum_{t=1}^{T}\sum_{m=1}^{t}\frac{\delta\mathcal{L}_c[t]}{\delta u_{out}[m]}\frac{\delta u_{out}[m]}{\delta a_l[m]}\frac{\delta a_l[m]}{\delta d_l} \tag{9}$$

In Equation (9), $\frac{\delta a_l}{\delta d_l}$ is equal to $\frac{da}{dt}$ for every unique synapse.

## 4 EXPERIMENTS

In this section we will elaborate on the experiments conducted for this study and the corresponding results. First we describe the datasets and the precise setup for training the SNNs. Following this, we present an analysis of the AdLIF neuron model and lastly show the full the results.

### 4.1 DATASETS

We used three common SNN benchmark speech recognition datasets: the Spiking Heidelberg Digits (SHD), the Spiking Speech Commands (SSC) (Cramer et al., 2020) and the Google Speech Commands v0.02 (GSC) dataset (Warden, 2018). The first two datasets are neuromorphic datasets, in which the original sounds were converted to spikes, spread out over 700 input channels. The GSC dataset consists of speech samples. The SHD dataset consists of German and English spoken digits (0 through 9). The SSC dataset is based on the sounds from the GSC dataset. Both the SSC and GSC datasets have 35 classes from a large group of speakers in a non-controlled environment. These larger datasets provide a more challenging speech recognition task.

Similarly to other works, for the spiking datasets, which were zero-padded and aligned to 1s, we binned the input data from 700 input channels to 140 channels and 100 timesteps with 10ms bins to ensure uniformity across all samples. Regarding the GSC dataset, speech data was aligned to 1s by padding with zeros and thereafter binned in 10 millisecond bins to generate samples of 100 timesteps. Further processing was performed with a Mel filterbank with 40 Mel filters. Since there is no predefined test set for the SHD dataset, we decided to take both a maximal and an average accuracy on the validation set across 10 experiments with different random seeds.

### 4.2 TRAINING SETUP

All neuronal trainable parameters were uniformly initialized between specific boundaries and subsequently co-learned with the other model trainable parameters to reflect the neuronal heterogeneity (Perez-Nieves et al., 2021). These parameters were initialized following a uniform distribution: $\alpha \in [0.36, 0.96]$, $\beta \in [0.96, 0.99]$, $a \in [0, 1]$ and $b \in [0, 2]$. Different from the AdLIF model (Bittar & Garner, 2022), we extended the available range of the membrane potential decay parameter $\alpha$ and limited the dependency of the adaptation current with respect to the membrane potential, $a$, to remain positive, which showed to stabilize the neuron model for sparse input data. As indicated earlier, we refer to our variant of this neuron model as AdLIF+. During the training process, all neuron parameters are clipped to remain within these boundaries. The spike threshold was fixed at 1 (dimensionless). The weights of all connections were initialised following the default Xavier uniform distribution and the neuronal biases were set to zero. For all hidden neurons, the initial membrane potential and adaptation current were randomly initialized, following a uniform distribution between 0 and 1.

We used the Adam optimizer (Kingma & Ba, 2014) in all experiments with an initial learning rate of 0.01 for the SHD dataset and 0.001 for the others, similarly to previous work (Bittar & Garner, 2022). The learning rate for the delays was always equal to $10 * lr_{weigths}$ We used a simple scheduler for both the weights and delays, which decreased the learning rate by a factor of 0.7 with a patience of 5 epochs. Dropout (Srivastava et al., 2014) was applied in the hidden layers with rates of 0.5 and 0.25 for SHD and the other datasets respectively. The available delay values are limited to remain within [0, 25] time steps for these datasets. The network is initialised without delays i.e. all delays are equal to zero and the input data was right padded accordingly for the network to allow delayed spikes. This exact setup was used for all experiments on all datasets presented in this work and implemented in PyTorch (Paszke et al., 2019). Our code is based on the Sparch implementation (Bittar & Garner, 2022).

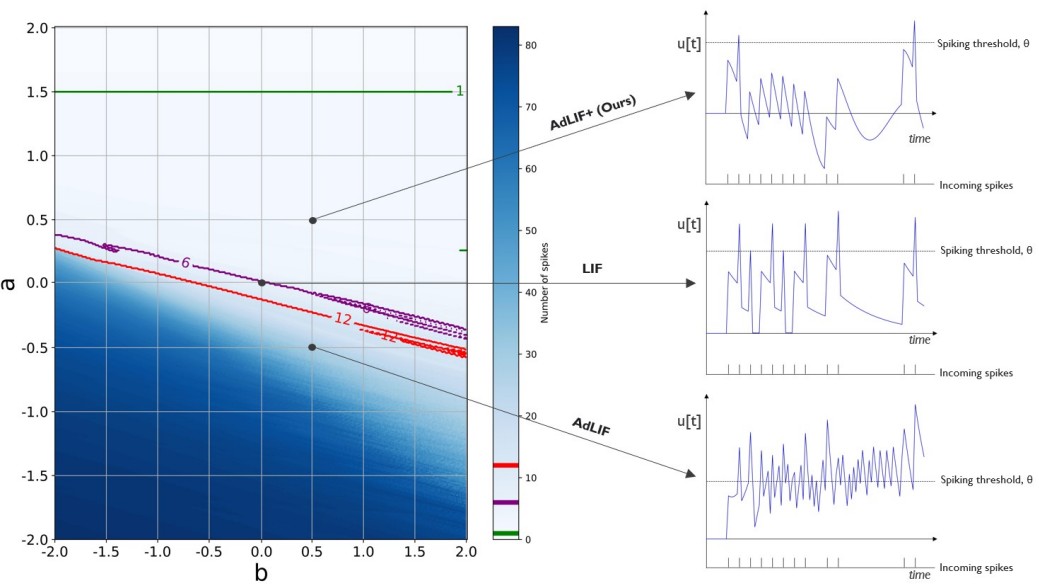

Figure 3: Illustration of different neuron model adaptation parametrizations and their responses to a fixed train of 12 incoming spikes. We analyse the number of **output** spikes and the corresponding evolution of the membrane potential u[t] over time for a range of $a$ and $b$ neuronal adaptation parameters. A model without adaptation, the LIF model, is found at where both $a$ and $b$ are equal to zero. The AdLIF model, which allows both positive and negative $a$, can possibly result in an unstable spiking regime. In this case the neuron generates more spikes than the number of incoming spikes, 12 in this example. We therefore limit the updated neuron model, AdLIF+ to remain within positive $a$ and $b$ boundaries. For reference, we also show the parametrizations for 6 and 1 output spikes in purple and green respectively.

In this section we analyse the effects of the different parameter ranges for the main adaptation parameters $a$ and $b$ from the AdLIF neuron model, as presented in Equation 1. Our AdLIF+ model differs from the AdLIF model in two ways: we extended the available decay window for the membrane potential, the $\alpha$ parameter, for the model to be able to forget more quickly if needed. More importantly, we limited the available range of the $a$ parameter, which allows the current membrane potential u[t] to influence the adaptation current w[t]. Similarly to adaptation by means of an adaptive threshold, this $a$ parameter mainly limits spikes if the membrane potential was high in the past and thus provides a homeostatic mechanism, given that this $a$ remains positive.

In Figure 3, we illustrate that for a possible negative $a$, the behavior of the adaptation current can lead to a non-controlled, chaotic spiking regime. In this figure we count the number of spikes that are produced by a neuron, given a single fixed input spike train of 12 spikes, for different adaptation parametrizations of $a$ and $b$. Whenever the produced number of spikes is higher than the number of input spikes, the model enters a chaotic regime in which a positive feedback loop could be activated and the neuron remains spiking, even without the presence of input spikes. Figure 3, shows that is mainly the case for a negative $a$, where apart from a region with a very high parameter $b$, the spike-triggered fraction of the adaptation current, the number of output spikes generated is very high. We therefore, unlike other works, preferred to remain in the top right quadrant with both positive $a$ and $b$, and hence the AdLIf+ name for our neuron model. In this figure, the base LIF model can be found at the point where $a$ and $b$ are equal to 0 and no adaptation current w[t] is produced.

## 4.4 RESULTS

In the first experiment we investigate the effects of adding the trainable parameters to a basic LIF neuron model with similar initialization, adding the trainable synaptic delays and the interplay of co-learning both as well as the effect of updated adaptation parametrization boundaries. The results in

terms of classification accuracy and highlighting the individual contributions are shown in Table 1, as tested on the SHD dataset. Due to its small size, the SHD dataset is more suitable for conducting exploratory experimemts. When comparing the LIF model with the AdLIF and the AdLIF+ model, independent of the inclusion of learning the delay value, we see that the average validation accuracy as well as its maximal value over 10 experiments is significantly increased by up to 7.2% and 9,6% respectively. The number of additional trainable parameters is very limited as it is only increased by about 2.1%. These results clearly show the added value of trainable adaptation parameters for SNN models. The AdLIF+ model performs about 2.5% better than the AdLIF model on average, which is in accordance with the empirical of adaptation parameters $a$ and $b$ in the analysis from section 4.3. These results show the importance of adequate trainable neuron parameter boundaries.

Table 1: Effect of the proposed modifications to a basic LIF-based 2 hidden layer feedforward SNN model with a hidden layer size of 128. Given the limited size of the SHD validation set, the (average, max) accuracy over 10 runs are shown.

|  | Basic LIF | AdLIF | **AdLIF+** | Synaptic delay LIF | **Synaptic delay AdLIF+** |
|---|---|---|---|---|---|
| # Parameters | 37.9k | 38.7k | 38.7k | 74.8k | 75.8k |
| Accuracy (%) | (84.49, 86.22) | (91.67, 93.85) | (94.19, 95.14) | (92.18, 93.02) | **(95.02, 96,26)** |

The addition of trainable synaptic delays further increases the accuracy for both the LIF and AdLIF+ neuron models. This shows that the extra provided capacity to utilize memory is beneficial for the speech recognition task, complementing the computational complexity provided by the trained adaptation. Logically, the number of trainable parameters is doubled as every synapse is now characterised by both a delay and a weight value.

An overview of the results of our experiments on the full ADLIF+ model with synaptic delays across all speech recognition datasets is presented in Table 2. We compared our SNN model to state-of-the-art SNN solutions from literature with a 2-hidden layer architecture and a corresponding state-of-the-art ANN model for each dataset, shown below the dotted line.

Table 2: Test Accuracy on SHD, SSC and GSC datasets and comparison with state-of-the-art in SNN for 2 hidden layer feedforward models.

| Dataset | Model | Hidden size | # Parameters | Accuracy |
|---|---|---|---|---|
| **SHD** | Adaptive RSNN (Yin et al., 2021) | 128 | / | 90.4% |
| | RadLIF (Bittar & Garner, 2022) | 1024 | 3.9M | 94.62% |
| | Axonal delays (Sun et al., 2022) | 128 | 0.1M | 92.36% |
| | Synaptic delays (Hammouamri et al., 2023) | 256 | 0.2M | 95.07±0.24% |
| | **AdLIF+, synaptic delays (ours)** | 128 | **0.076M** | **96.26%** |
| | CNN (Cramer et al., 2020) | / | / | 92.4% |
| **SSC** | Adaptive RSNN (Yin et al., 2021) | 400 | / | 74.2% |
| | RadLIF (Bittar & Garner, 2022) | 1024 | 3.9M | 77.4% |
| | 1D-Conv synaptic delays (Hammouamri et al., 2023) | 512 | 0.7M | 79.77 ± 0.09% |
| | **AdLIF+, synaptic delays (ours)** | 512 | **0.71M** | **79.81%** |
| | GRU (Bittar & Garner, 2022) | 512 | / | 79.05% |
| **GSC** | RSNN, LIF (Zenke & Vogels, 2021) | 256 | / | 85.3% |
| | RSNN with SFA (Salaj et al., 2021) | 2048* | / | 88.5% |
| | RadLIF (Bittar & Garner, 2022) | 512 | 0.83M | 94.51% |
| | Synaptic delays (Hammouamri et al., 2023) | 512 | 0.7M | 94.91 ±0.09% |
| | **AdLIF+, synaptic delays (ours)** | 512 | **0.61M** | **95.38%** |
| | GRU (Bittar & Garner, 2022) | 512 | / | 94.32% |
| | Transformer (Gong et al., 2021) | / | / | 98.11% |

*Recurrent SNN with a single hidden layer.

For the SHD dataset, the AdLIF+ model with trained synaptic delays matches the state-of-the-art results of a recently proposed alternative method for training synaptic delays at just a fraction (less than half) of its number of trainable parameters and outperforms all other SNN and ANN methods on this dataset. On the harder SSC and GSC datasets, the AdLIF+ model with trainable synaptic delays shows on par or better performance than the current state-of-the-art models in SNN with

a similar number of trainable parameters or less. Furthermore, one additional SNN model with 3 hidden layers was proposed Hammouamri et al. (2023). This model slightly outperforms ours on the SSC dataset 80.29 ± 0.06% with an increase of about 45% additional trainable parameters, but only achieved 95.29 ± 0.11% on the GSC dataset, which is similar to our SNN with just 2 hidden layers. Given the budget of trainable parameters, the proposed feed-forward SNN model even outperforms a non-SNN recurrent model (GRU) with the same preprocessing and moves closer to the performance of a large ANN model with transformer architecture.

## 5 Conclusion

In this paper, we presented a novel SNN model, the AdLIF+ with trainable synaptic delays. To the best of our knowledge, this is the first SNN model in which the synaptic delays are directly learned in coordination with the neuronal adaptation. Furthermore, we showed that 1) it is possible to co-learn synaptic weights, delays and neuronal adaptation parameters at the same time, and 2) co-learning these parameters proved to mutually benefit the optimization of all learned parameters as shown for three speech recognition datasets. We highlighted the importance of co-learning the synaptic delays and neuronal adaptation parameters with the synaptic weights and showed that they mutually benefit from this co-optimization, reaching state-of-the-art performance in SNN on all investigated datasets. The superior performance can be attributed to two additional features: 1) Training the synaptic delays enables a neuron in the SNN to explicitly correlate temporally distanced features 2) the trained neuronal adaptation allows a greater variety in spike patterns, widening the feature space to be explored.

We showed that for a very simple architecture, a 2 hidden layer fully connected feedforward network, we are able to compete against or outperform larger ANN models, with a limited number of additional trainable SNN parameters. The performance of the presented SNN model shows the promise of SNN research on tasks with rich temporal dynamics, and in particular research on biologically inspired extensions to existing SNN models.

When comparing to larger ANN models, the performance of the presented SNN model is lacking. A future step in our research is therefore to investigate how learning delays and adaptation parameters are influenced by the model architecture. More advanced architectures such as convolutional spiking neural networks or experimenting with the training recurrent synapses could further bridge the gap with ANNs. Another exciting avenue to be explored is the chosen neuron model. As the proposed AdLIF+ model is still one particular Generalized integrate-and-fire model version Gerstner & Kistler (2002) and Gerstner et al. (2014) and many more exist, there are plenty available neuron model extensions to be investigated. Future research will point out to what extent these will be useful for application in SNN in the context of 1) their additional performance in terms of classification accuracy and 2) the additional complexity for their deployment on dedicated neuromorphic hardware.

Another point to explore is that our work includes a fixed maximal delay, which needs to be defined before training and requires fine-tuning. In future work we intend to investigate the effect of co-learning the synaptic weights and delays on more complex neuron models and model architectures as well as validate them on non-sound datasets.

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

## A APPENDIX

### A.1 TRAINING HYPERPARAMETERS

Table 3 shows an overview of the exact parametetrization of the experiments, executed for this study. The setups for the SSC and GSC datasets were identical.

Table 3: Overview of the hyperparameters used in training for all datasets considered in this study

| Dataset | Hidden layer size | Epochs | Batch size | Dropout rate | Lr weights | Lr delays | Initialization | Delay caps |
|---------|-------------------|--------|------------|--------------|------------|-----------|----------------|------------|
| SHD | 128 | 100 | 128 | 0.5 | 0.01 | 0.1 | Xavier uniform | [0,25] |
| SSC & GSC | 512 | 200 | 32 | 0.25 | 0.001 | 0.01 | Xavier uniform | [0,25] |

One of the properties that affects the efficiency of the SNN model, when deployed on neuromorphic hardware is the effective number of spikes that are required when processing a single sample. We therefore analyse the number of spikes in the hidden layers to assess how the proposed improvements influence the total number of spikes in the SNN. To account for inter-experimental variance, we averaged the spike rates over 10 runs on the SHD dataset. The results are shown in in Table 4.

We see that the number of spikes is not significantly increased by the addition of any of the considered adaptation methods. The synaptic delays however result in an increase of the number of spikes in the SNN model.

| Model | Basic LIF | AdLIF | **AdLIF+** | Synaptic delay LIF | **Synaptic delay AdLIF+** |
|---|---|---|---|---|---|
| Number of spikes/neuron | 10.75 | 10.7 | 10.28 | 13.7 | 11.48 |

Table 4: The average number of spikes per neuron for a single sample, compared over different SNN models. These results were averaged over 10 runs on the SHD dataset.

## A.3  MODEL ARCHITECTURE

Figure 4 shows the network architecture. The input spikes are passed trough 2 hidden layers. In the readout layer the membrane potential is extracted from memoryless output neurons. The recurrent connections represent the membrane potential leakage and the adaptation currents.

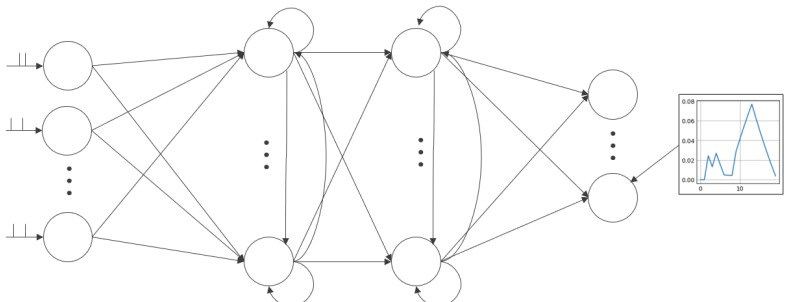

Figure 4: Network architecture: a 2-hidden layer feedforward SNN model

## A.4  DETAILED RESULTS ON SHD DATASET

In this section we show the results of the 10 different runs with different pseudo-random seeds. Figure 5 provides a more general overview than the average and maximal results, as provided in Table 1. These results again show the importance of both the neuronal adaptation and the trainable synaptic delays in the SNN model recognition performance.

## A.5  VISUALISATION: TRAINED NEURONAL PARAMETERS

In this section we visualize the distribution of the trained parameters for every layer in the fully connected SNN, trained on the SHD dataset. This particular model achieved an accuracy of 95.45% on the validation set with the displayed parameter distribution.

We observe that although all neuronal parameters were initialized uniformly between their respective boundaries: $a \in [0, 1]$ and $b \in [0, 2]$, the optimized neuronal parameters follow a different distribution. Mainly parameter $b$ shows a remarkable results. All other distributions are spread relatively evenly between their boundaries. We see that apart from some neurons, most neurons converged to a $b$ value of zero. This suggests that the parameter linking the adaptation current with the sub-threshold membrane potential, $a$, is predominantly used for adaptation. This is in accordance with our empirical results, shown in Figure 3, where mainly a influences the total number of output spikes for a given input spike train.

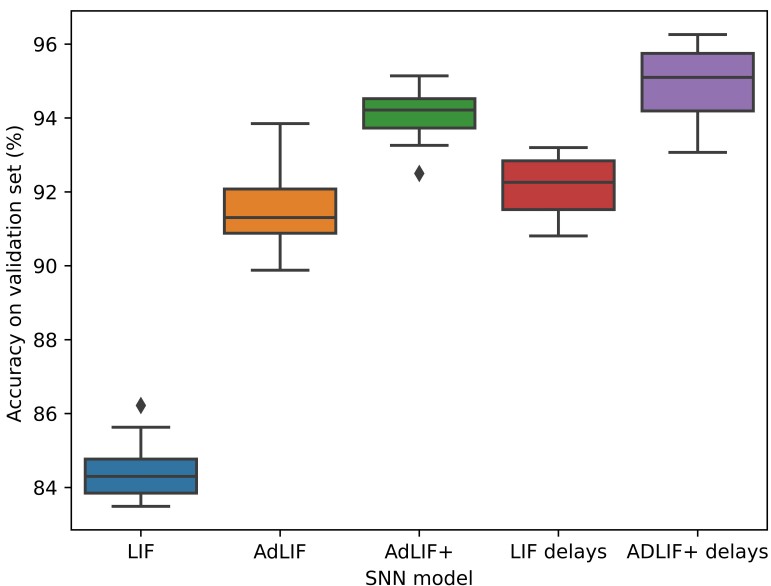

Figure 5: Detailed overview of the results on the SHD dataset for all SNN model setups.

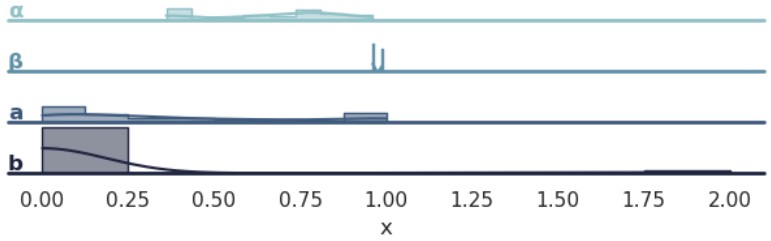

Figure 6: Distributions of the trained parameters ($\alpha$, $\beta$, a and b) in the first hidden layer

## A.6 VISUALISATION: TRAINED DELAYS

In our experiments, the delays were initialized at zero, as if there was no delay in the SNN. In Figure 8, we show the final delay distribution of every layer in the SNN for the same checkpoint as shown in section (A.5) for the neuronal parameters (validation accuracy at 95.45%). For the input-

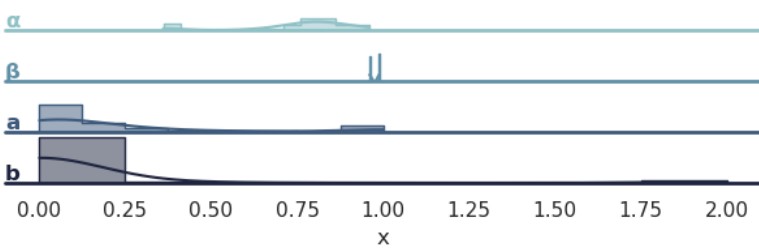

Figure 7: Distributions of the trained parameters ($\alpha$, $\beta$, a and b) in the second hidden layer

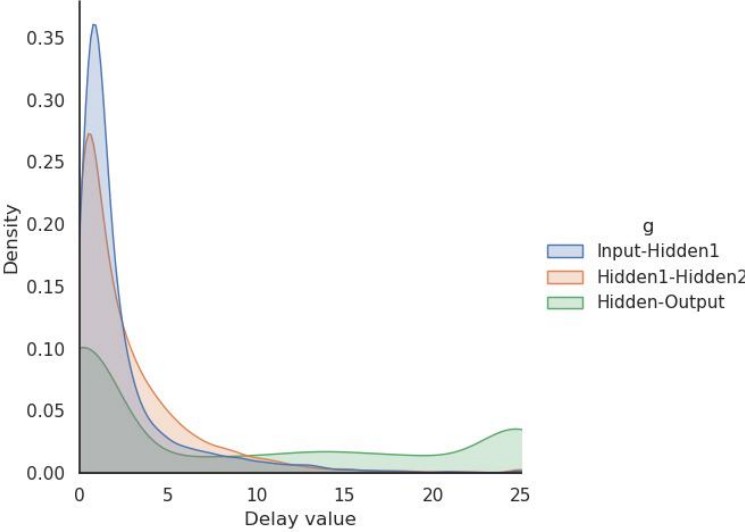

Figure 8: Distributions of the trained delays in the 3 trained layers of the SNN model

Hidden1 and Hidden1-Hidden2 layers, we see that the optimized delays are centered around 0 and 1 while in the output layer this spread is wider. We hypothesize that in the first 2 layers, short-term dependencies are prioritized over longer term ones.

## A.7  ON SPIKING NEURON MODELS

Over the past few years, neuroscientists have proposed many neuron models Gerstner & Kistler (2002). On one side, the Hodgkin-Huxley model, characterised by four differential equations, can accurately reproduce electrophysiological measurements from biological neurons. The biological accuracy comes at the cost of intrinsic complexity, limiting their use in simulations of large networks of neurons. On the other side, the Leaky integrate-and-fire (LIF) neuron, characterised by one differential equation, was proposed for efficient processing and capturing the core properties of spiking neurons. Nevertheles, it is known that LIF neurons are not sufficient to describe the diversity of responses to input patterns found in neurons.

The AdLIF+ neuron, characterised by two differential equations, proposed in this work, finds the balance between both ends. As shown in Gerstner et al. (2014) the AdLIF+ model is able to reflect a wide range firing patterns. In the biophysical interpretation, the sub-threshold adaptation parameter $a$ can be linked to the slower dynamics of ion channels in relation to the membrane potential. The spike-triggered adaptation parameter $b$ is linked with ion channels only opening when the membrane potential is above the spike threshold.

There are many more biologically inspired additions we can make to the proposed AdLIF+ model in order to improve its processing power. Non-linear (exponential) or quadratic (Izhikevich) neurons, multicompartment models, refractoriness and stochastic neurons are just a few examples of possible enhancements. Future research should point out to what extent these modifications are also useful for applications with SNN. Potential gains in SNN model performance should be weighted with likely, additional complications in the deployment of complex neuron models on dedicated neuromorphic hardware.

