# OpenReview forum: "Co-learning synaptic delays, weights and adaptation in spiking neural networks"
_ICLR.cc/2024/Conference — ICLR 2024 Conference Withdrawn Submission_

### Official Review · Reviewer_mMin · 2023-10-31

**Soundness:** 2 fair
**Presentation:** 2 fair
**Contribution:** 2 fair
**Rating:** 5
**Confidence:** 5

**Summary:**

This paper introduce two biologically plausible features in spiking neuron networks:
1.  On the basis of existing AdLIF spiking neurons, a set of parameters are introduced to better describe the adaptive process of neurons, and achieve a steady-state behavior during it’s spiking process by limiting the boundaries of some trainable parameters.
2.  Introduce the synaptic propagation delay in the network, which enhances the SNN’s ability to process temporal information.

The SNN incorporating these two features exhibit richer dynamic in the temporal dimension comparing to common LIF node. And through some analysis and experiments on three language datasets. This work verified the superiority of this novel SNN. The performance shows the promise of SNN research on tasks with rich temporal dynamics, and in particular research on biologically inspired extensions to existing SNN models.

**Strengths:**

* A detailed introduction is given to the neural dynamics and operating mechanism of the SNN proposed in this paper.

The article provides detailed mathematical formulas to describe the operation process of the pulse neural network, with effective explanations for all parameters.

* The inspiration sources and effects of the new features introduce in the proposed SNN were analyzed and explained through effective experiments.

The article explains the sources of each new feature introduced, and provides paper analysis and specific experiments to demonstrate the benefits of the new features provided. At the same time, the effectiveness of the model was verified through specific experiments on three speech recognition datasets, and the comparison method cited is indeed relatively new.

**Weaknesses:**

* There are some slight discomfort in writing.

In section 3.2.1 and 4.1, there are sentences like “Similarly to earlier studies / other works”, but these “earlier studies” are not closely followed by the relevant references (though they may be papers that have already been cited earlier). And in the experimental results shown in the Table. 1, It seems that the 96.26 in row <Synaptic delay AdLIF+> is mistakenly written as ’96, 26’.

* There is a slight lack of completeness in the experimental results

The AdLIF+ neuron in this paper is modified based on the AdLIF neuron after a certain degree of parameter constraints. Would it be better to supplement the experimental results of “Synaptic delay AdLIF” in the first experiment on SHD (as shown in Table 1)? What’s more, I find that the accuracy of the proposed SNN on SHD in Table 2 is consistent with the so-called “maximal accuracy” in Table 1. Does this mean that the shown experimental results conducted on SHD are all the highest accuracy among 10 runs? Adopting such experimental results seems a bit imprecise if so (of course, the average accuracy in Table 1 is already high enough. The main concern is the clarity of the results).

**Questions:**

1. Please refer to the above weakness.
2. What about the performance of the proposed models when applied into the larger dataset?
3. Is there any advantage on the training speed for the proposed model compared with other methods?

**Details Of Ethics Concerns:**

None.

---

### Official Review · Reviewer_fvm6 · 2023-10-31

**Soundness:** 3 good
**Presentation:** 3 good
**Contribution:** 2 fair
**Rating:** 3
**Confidence:** 4

**Summary:**

The paper presents a novel Spiking Neural Network (SNN) model, the AdLIF+, in which synaptic weights, adaptation parameters and delays are co-optimized. This heterogeneity leads to improved performance leading to the SNN outperforming an ANN on Neuromorphic datasets.

**Strengths:**

- The paper is clear and easy to read
- Heterogeneity is an important research direction in SNNs and this paper demonstrates that on multiple datasets

**Weaknesses:**

- There have been previous works that have optimized over neuronal parameters to introduce heterogeneity, thus I feel the novelty of the work is quite limited.
- Datasets are too simple and the advantages of the proposed approach might not be representative

**Questions:**

- Could you try scaling the size of the model and complexity of the dataset
- what happens if you optimize the adaptation and delays separately like it is done in https://openreview.net/pdf?id=bp-LJ4y_XC
- In section 4.4, when you discuss the increase in number of trainable parameters, I think it would be better to discuss it as a function of size: Number of Neurons (N) and Number of weights (W) rather than pure numbers for a single size.

---

### Official Review · Reviewer_XBEc · 2023-11-01

**Soundness:** 1 poor
**Presentation:** 2 fair
**Contribution:** 3 good
**Rating:** 3
**Confidence:** 5

**Summary:**

This article has improved the information transmission process of traditional LIF neurons and has incorporated the concept of delay, training it together with the weights. The author investigated its effects in a feedforward network with two hidden layers across three speech datasets, demonstrating its superiority. From a bio-interpretability perspective, this model enriches the heterogeneity and dynamic characteristics of LIF neurons, making it very interesting.

**Strengths:**

1. The author's idea of incorporating delay and membrane potential updating into the surrogate gradient training process, enabling the SNN to exhibit more heterogeneity, is intriguing. Especially noteworthy is that there are currently few studies that combine mainstream surrogate gradient methods with biological features like synaptic delay.
2. Figure 3 is particularly fascinating as it explains how parameters influence the dynamic properties of neurons.

**Weaknesses:**

1. As I mentioned in the advantages, integrating these biological characteristics into the surrogate gradient learning process is interesting. However, what motivates this integration? The author seems not to have explained this clearly. As mentioned in the introduction, if the goal is solely to enhance biological interpretability, could we achieve better optimization results by learning more parameters?

2. Eq 1 and 7 are crucial to this paper. The author should use the formulas to analyze the problems with the current model, including insufficient biological interpretability, from a theoretical perspective. Then, introducing these formulas would better help readers understand the author's motivation.

3. Furthermore, I think it is necessary to add more information about how each parameter affects the network. This could be done by experimentally explaining the benefits of adding these parameters and showing that they can be effectively optimized through BPTT. As illustrated in Fig 3, there should be additional information on how different response modes specifically affect model performance.

4. Moreover, the impact of different models and depths on the proposed method should be supplemented. Additionally, conducting experiments on a broader range of datasets would better support the author's claim of comprehensive improvement.

5. Despite the introduction of the intermediary variable w, as shown in Eq 5, the model still cannot overcome the problem of gradient vanishing over time during the optimization process, which ensures its effectiveness in optimizing temporal information.

6. Other issues: It is recommended to change the variable w to another parameter, as it is more commonly used to represent weights.

**Questions:**

1. Some issues are as indicated in the weakness section.
2. While the author has enhanced the performance of SNNs by adding several parameters, the increase in the number of parameters is substantial. Has there been any consideration for a comparison with RNNs or GRUs or LSTM under the same parameter quantity conditions?
3. Furthermore, after adding so many parameters, how does the loss curve of model optimization change? Is there a possibility of collapse? Therefore, has there been any consideration regarding the selection of parameter boundaries?

---

### Official Review · Reviewer_2Gfg · 2023-11-02

**Soundness:** 2 fair
**Presentation:** 2 fair
**Contribution:** 2 fair
**Rating:** 3
**Confidence:** 4

**Summary:**

The authors proposed a new learning algorithm for the SNNs in which both synaptic weights and delays are co-optimized in collaboration with the neuronal adaptation parameters. Various experiments are conducted to verify its performance.

**Strengths:**

1、 Compared to the existing works, this work shows certain advantages in accuracy.

**Weaknesses:**

1、The novelty of this work may be limited as there have been numerous studies that have already explored the plasticity of synaptic delays.

2、This work has not been compared to mainstream learning algorithms for Spiking Neural Networks (SNN) using popular datasets such as TET, DVS-CIFAR10, and DVS-Gesture, among others.

3、The figures presented in this manuscript are somewhat rough in terms of their quality and level of detail

**Questions:**

see weaknesses

---

### Author Response · Authors · 2023-11-22

We want to thank all reviewers for their time and constructive feedback. We are happy that most of the reviewers recognized the importance of heterogeneity in learning SNN through neuronal adaptation and synaptic delays via surrogate gradients. The main novelty we proposed is a mechanism for co-learning weight and delay since most research focuses on learning techniques that decouple them. However, as pointed out and required by the reviewers, a more comprehensive comparison with mainstream Spiking Neural Networks learning algorithms and model architectures is beyond the available time frame for this submission and, in fact, out of the scope of this work. We, therefore, decided to withdraw our submission. Once again, we appreciate the time and comments from the reviewers.